# Review on Characterization of Biochar Derived from Biomass Pyrolysis via Reactive Molecular Dynamics Simulations

**Zhong Hu** [1],* and **Lin Wei** [2]

1    Mechanical Engineering Department, South Dakota State University, Brookings, SD 57007, USA
2    Agricultural and Biosystems Engineering Department, South Dakota State University,
     Brookings, SD 57007, USA; lin.wei@sdstate.edu
*    Correspondence: zhong.hu@sdstate.edu; Tel.: +1-605-688-4817

**Abstract:** Biochar is a carbon-rich solid produced during the thermochemical processes of various biomass feedstocks. As a low-cost and environmentally friendly material, biochar has multiple significant advantages and potentials, and it can replace more expensive synthetic carbon materials for many applications in nanocomposites, energy storage, sensors, and biosensors. Due to biomass feedstock species, reactor types, operating conditions, and the interaction between different factors, the compositions, structure and function, and physicochemical properties of the biochar may vary greatly, traditional trial-and-error experimental approaches are time consuming, expensive, and sometimes impossible. Computer simulations, such as molecular dynamics (MD) simulations, are an alternative and powerful method for characterizing materials. Biomass pyrolysis is one of the most common processes to produce biochar. Since pyrolysis of decomposing biomass into biochar is based on the bond-order chemical reactions (the breakage and formation of bonds during carbonization reactions), an advanced reactive force field (ReaxFF)-based MD method is especially effective in simulating and/or analyzing the biomass pyrolysis process. This paper reviewed the fundamentals of the ReaxFF method and previous research on the characterization of biochar physicochemical properties and the biomass pyrolysis process via MD simulations based on ReaxFF. ReaxFF implicitly describes chemical bonds without requiring quantum mechanics calculations to disclose the complex reaction mechanisms at the nano/micro scale, thereby gaining insight into the carbonization reactions during the biomass pyrolysis process. The biomass pyrolysis and its carbonization reactions, including the reactivity of the major components of biomass, such as cellulose, lignin, and hemicellulose, were discussed. Potential applications of ReaxFF MD were also briefly discussed. MD simulations based on ReaxFF can be an effective method to understand the mechanisms of chemical reactions and to predict and/or improve the structure, functionality, and physicochemical properties of the products.

**Keywords:** biochar; biomass; pyrolysis; carbonization; physicochemical property; thermochemical; molecular dynamics (MD) simulation; reactive force field (ReaxFF)

## 1. Introduction

Biochar is a carbon-rich solid material that can be produced by oxygen-limited thermochemical processes (e.g., torrefaction, pyrolysis, gasification, combustion, etc.) [1–6]. Many types of organic materials, including agricultural residues and forest solid waste, can be used as feedstock for biochar production. This biochar usually contains carbon (C) (40–90%), oxygen (O) (1–40%), hydrogen (H) (1–10%), nitrogen (N) (0–5%), sulfur (S) (<1%), and other trace elements [7–9]. Depending on biomass feedstock species, reactor types, and operating conditions, such as particle size, heating rate, residence time, carrier gas, reaction temperature, pressure, etc., the proportions and physicochemical properties of different compounds of biochar may vary greatly. Biochar has various significant advantages and potentials due to its remarkable properties [10–13]. For thousands of years, it has been used as a soil amendment to improve agricultural productivity, but recently, the use of biochar



as a precursor, which is activated or functionalized into carbon materials, such as graphite, graphene, carbon black, or synthetic carbon fibers (CFs), carbon nanofibers (CNFs), carbon nanotubes (CNTs), carbon nanospheres (CNSs), etc., has become more and more attractive for many applications in nanocomposites, energy storage, sensors, biosensors, etc. Carbon materials produced from non-renewable sources such as coal, oil, and natural gas may produce excessive greenhouse gas (GHG) emissions, accelerate the consumption of fossil fuel resources, and are not economically viable. In contrast, biomass resources are abundant, available, renewable, low-cost, harmless, high in carbon content, and low in sulfur and ash [10–17]. In addition, applications of biochar can promote carbon sequestration and reduce $CO_2$ emissions into the atmosphere, improving resilience to global climate change [10,15]. Cancer-targeted drug delivery systems based on carbon nanostructures are another promising application due to their ability to selectively recognize specific receptors overexpressed in cancer cells [18,19]. Therefore, various methods, such as laser ablation, gamma irradiation, chemical vapor deposition (CVD), arc-discharge, catalytic chemical vapor deposition (CCVD), hydrothermal carbonization [18,19], plasma technique [8,20], and plasma-enhanced chemical vapor deposition (PECVD), have been developed to synthesize different carbon nanomaterials (e.g., CNTs) [14]. Thus, the biochar market is expected to reach USD 3.14 billion by 2025 [14,15,21,22].

Biochar is a complex carbonaceous material with many physical and chemical parameters that control its reactivity towards inorganic and organic substances in aqueous solution [23]. Some chemical properties of biochar vary with the type of parent biomass feedstock species, with the most pronounced and consistent changes in volume and surface chemistry occurring with production conditions, biochar surface charge and ion exchange capacity, environmental influences, etc. [24]. Biochar has also attracted considerable attention as a filler material in polymer matrices and as a reinforcement. The fabrication of bio-composites employs a variety of methods. The effect of adding biochar to the overall composites shows great promise in improving the performance of the overall composites [25]. Various modification methods were proposed to enhance certain functions of biochar. However, these modifications may also lead to structural uncertainty, additional energy consumption, secondary pollution, and/or additional costs. The most commonly used biochar modification methods can be classified according to the purposes of modification, such as surface area enlargement, persistent free radical manipulation, magnetism introduction, and redox potential enhancement. More importantly, the balance considerations of biochar designs, such as the balance between effectiveness and stability, function and risk, effectiveness and cost, etc., were systematically analyzed and discussed [26,27]. To determine the synergistic effects of biomass feedstock species, reactor types, operating conditions, and post-treatments on biochar activation and functionalization, the surface chemistry, physiochemistry, structural, and molecular characterization of the produced biochar, as well as the relationships between process parameters and biochar surface morphology and internal microstructure, have been intensively studied since the last few decades [28–30]. The surface chemistry, structural, and molecular characterization of the produced biochar have been investigated using various advanced characterization techniques. For example, the chemical characterization of biochar can be tested using elemental analysis, Fourier transform infrared spectroscopy (FTIR), X-ray photoelectron spectroscopy (XPS), and scanning electron microscope—energy dispersive spectrometer (SEM-EDS), as well as statistical analysis of experimental data [31]. Regarding surface and structural characterization, biochar and activated carbons can have varying degrees of aromaticity, which generally refers to the level of defects and the size of the aromatic rings in the bulk material. Since micropores formed by stacking faults between layers of aromatic ring clusters in porous carbons are responsible for the high effective surface area in these materials, it is crucial to characterize these phases formed of $sp^2$ hybridized carbons. Raman spectroscopy (Raman), scanning electron microscopy (SEM), specific surface area and pore size analyzer, high-resolution transmission electron microscopy (HR-TEM), X-ray diffraction (XRD), XPS, etc., provide simple and non-destructive means to achieve this

goal, and diffuse reflectance infrared Fourier transform spectroscopy (DRIFTS) can be used semi-quantitatively [32,33]. To characterize the structure of biochar at the molecular level, HR-TEM, Raman, XPS, etc., can be used [30,34]. The 3D morphology and distribution of biochar pores can be studied by using a density analyzer, X-ray computed tomography (XCT), and SEM [35–37]. In summary, many modern characterization techniques have been reported for the characterization of biochar, such as SEM, FTIR, XRD, thermogravimetric analysis (TGA), nuclear magnetic resonance spectroscopy (NMR), Brunauer–Emmett–Teller (BET), Raman, density analyzer, proximate and ultimate analysis, etc. [38,39].

Biochar can be produced from biomass using various methods. Biochar physicochemical properties, including chemical composition, carbon content, ash content, surface area and morphology, pore size and distribution, functionality, etc., may be affected by the biomass feedstock species nature, pretreatment, pyrolysis conditions (such as reaction temperature, heating rate, retention time), functionalization strategies (e.g., magnetic biochar, plasticized biochar, and co-composed biochar), and so on. The physical chemistry of biochar functioning includes the thermodynamics of biochar adsorption/desorption, the kinetics of biochar adsorption/desorption, the meaning of reaction order and Langmuir isotherm from kinetic considerations, the dynamics of water and nutrients in the biochar pore system, and the trapping and decomposition mechanisms of pollutants in biochar [40–43]. However, understanding biochar seems more experimental and empirical than relying on a well-structured theoretical framework. From previous studies, the properties and functionalities of biochar depend on many factors, including the composition, morphology, size, internal structure, pore size and distribution of biochar clusters, processing methods, etc. However, there is still a lack of quantitative analysis and a comprehensive understanding of the effects of these factors and their interactions on the physicochemical properties and functionalities of biochar, which are critical for producing high-quality biochar as a precursor to effectively developing innovative carbon materials for new applications. The effects of these factors are usually not in a monotonically changing pattern (monotonically increasing or monotonically decreasing). There is usually an inflection point (threshold) where the material performance is at a maximum or minimum point, which can be fully considered when conducting biochar production and effective new/novel carbon material designs.

Depending on the problem and the spatial and time scales of interest, various computer modeling-based approaches to materials design/research exist, ranging from quantum mechanics (QM) to continuum simulations. Molecular dynamics (MD) or first-principles simulations are ideal for studying the properties of materials at the nanoscale. MD is an atomistic-scale simulation that describes the interactions between atoms through interatomic potentials. In the MD method, electronic effects are averaged, and the time evolution of atomic positions and velocities is calculated according to Newton's equations of motion. The electron correlation approximation is based on the Born–Oppenheimer theory that the MD time step used to describe the atomic motion is sufficient for electrons to achieve their ground stable state compared to the nuclei due to the difference in mass. Interatomic potentials (force fields) are established from the first principles or experimentally to describe the interactions between the atoms, including the effect of electrons, in terms of reproducible forces. The reliability of the interatomic potential determines the accuracy of the MD simulations and is also related to the ability to bridge the mesoscale methods [44–49]. With the development of MD in the past few decades, MD has developed into a ubiquitous, versatile, and powerful computational method for basic scientific research in biology, chemistry, biomedicine, physics, etc. [50]. Driven by the rapid development of supercomputing technology in recent decades, MD has entered the field of engineering as a first-principles predictive method for material properties, physicochemical processes, and even as a design tool. These developments have far-reaching implications and are discussed in recent papers focusing on MD of combustion and energy systems, gas/liquid/solid fuel oxidation, pyrolysis, catalytic combustion, electrochemistry, nanoparticle synthesis, heat transfer, phase change, and fluid mechanics. Due to the practical availability of MD

simulations of large-scale reactive chemical systems, the reactive force field (ReaxFF) system was developed and successfully applied to biomass systems [29,30,50–59]. ReaxFF uses a general relationship between chemical bond distance and bond order to separate atoms. The other valence terms (angular and torsional) present in the force field are defined in terms of the same bond orders so that all these terms smoothly go to zero when the bonds break. In addition, ReaxFF has Coulomb and Morse (van der Waals) potentials to describe non-bond interactions between all atoms. These non-bond interactions are shielded at short distances, so Coulomb and van der Waals interactions become constant as the distances approach zero. The parameters of the ReaxFF were derived from quantum chemistry (QC) calculations on bond dissociation and reactions of small molecules, as well as heat of formation and geometric data for some stable hydrocarbons [51].

The current theoretical framework of MD methodology covers both classical and reactive MD. The ReaxFF MD can simulate chemical reactions such as carbonization in biomass pyrolysis within the MD framework using QC calculations and/or force field representations of experimental data. Therefore, this paper reviews the fundamentals of the ReaxFF methods and MD simulations based on ReaxFF MD to characterize the physicochemical properties and biomass pyrolysis process. The simulation of the carbonization reaction in biomass pyrolysis, including the reactivity of the three major components: cellulose, lignin, and hemicellulose, was discussed. The potential applications of ReaxFF MD were also briefly discussed.

## 2. Fundamentals of ReaxFF MD Simulations

Modeling atomic and molecular systems requires computationally intensive QM methods such as, but not limited to, QM-based ab initio calculations [60] or density functional theory (DFT) [46,48,58,61–63]. These methods have successfully predicted various properties of atomic-scale chemical systems [53]. Due to the inherent non-locality of QM, the scalability of these methods ranges from $O(N^3)$ to $O(N^7)$, depending on the method used and the approximations involved. This greatly limits the size of simulated systems to a few thousand atoms, even on massively parallel platforms. Furthermore, the scaling of the computational cost as a function of the system size (~$O(N^3)$ for the DFT methods and ~$O(N)$ for the ReaxFF method) still limits the possibility of running QM or ReaxFF simulations long enough to produce large carbonization structures [64]. On the other hand, classical approximations of quantum systems, while computationally (relatively) easy to implement, produce simpler models that lack fundamental chemical properties such as reactivity and charge transfer. The recent work by van Duin et al. [51–64] carefully incorporated limited non-locality (to simulate quantum behavior) via empirical bond order potentials, overcoming the limitations of non-reactive classical MD approximations while largely preserving the computational simplicity of classical MD. The reactive system force field, ReaxFF, was developed. ReaxFF uses a general relationship between bond distances and bond orders on the one hand, and bond orders and bond energies on the other, resulting in the proper dissociation of bonds into separated atoms. The other valence terms (angular and torsional) present in the force field are defined in terms of the same bond orders so that all these terms smoothly go to zero when the bonds break. In addition, ReaxFF has Coulomb and Morse (van der Waals) potentials to describe all non-bonded interactions between all atoms.

### 2.1. General Force Field

The ReaxFF interatomic potentials are powerful computational tools for exploring, developing, and optimizing material properties. Although methods based on QM principles provide valuable theoretical guidance at the electronic level, they are often too computationally intensive for simulations that consider the complete dynamic evolution of a system. Like empirical non-reactive force fields, reactive force fields divide the system's energy into various partial energy contributions. Russo and van Duin [53] described the ReaxFF potential forms in detail. Therefore, only a brief overview of the central concepts of the method is presented here. ReaxFF uses a bond-order form combined with a description of

polarizable charges between atoms. This enables ReaxFF to accurately model the covalent and electrostatic interactions of various materials, as shown in Figure 1.

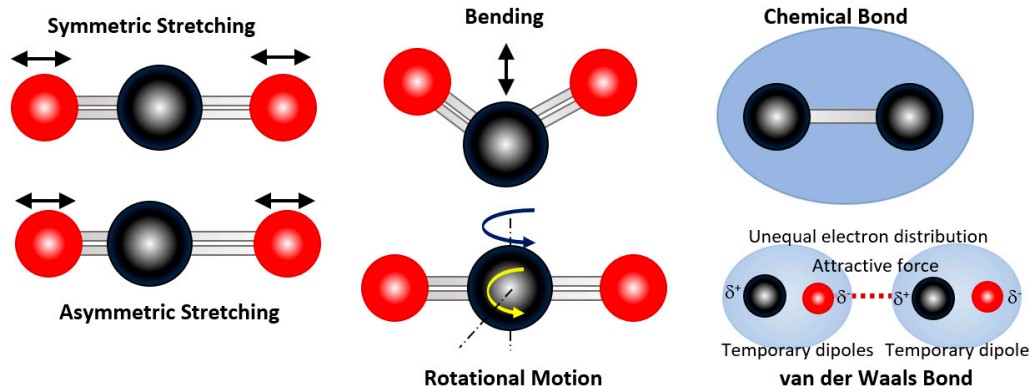

**Figure 1.** Some bond orders considered in ReaxFF potential forms.

The energy contributions to the ReaxFF potentials are summarized below [49,51,55,57,58]:

$$E_{\text{system}} = E_{\text{bond}} + E_{\text{over}} + E_{\text{angle}} + E_{\text{tors}} + E_{\text{vdWaals}} + E_{\text{Coulomb}} + E_{\text{specific}} \tag{1}$$

where the contributions to the total system energy, $E_{\text{system}}$, are the atomic bonds (continuous functions of interatomic distances, describing the energies associated with forming bonds between atoms), atomic over-coordination penalties (energy penalties that preventing atoms from over-coordination, based on valence rules, e.g., stiff energy penalties are applied if carbon atoms form more than four bonds), atomic valence angle and torsion (the energies related to three-body valence angle strain and four-body torsional angle strain), atomic non-bonding Coulombic and van der Waals energies (electrostatic and dispersive contributions calculated between all atoms, regardless of connectivity and bond-order), and the atomic specific term (system specific term that is generally not included unless required to capture properties particular to the system of interest, such as atomic lone-pairs, atomic conjugation, hydrogen binding, and $C_2$ corrections).

### 2.2. Bond Order and Bond Energy

A basic assumption of ReaxFF is that the bond order $BO_{ij}$ between atomic pairs can be obtained directly from the potential energy, which is divided into bond-order-dependent and bond-order-independent contributions. The bond order is calculated directly from the interatomic distances using an empirical formula:

$$BO_{ij} = BO_{ij}^{\pi} + BO_{ij}^{\pi\pi} = exp\left\{p_{bo,1}\left(\frac{r_{ij}}{r_o^{\sigma}}\right)^{p_{bo,2}}\right\} + exp\left\{p_{bo,3}\left(\frac{r_{ij}}{r_o^{\pi}}\right)^{p_{bo,4}}\right\} + exp\left\{p_{bo,5}\left(\frac{r_{ij}}{r_o^{\pi\pi}}\right)^{p_{bo,6}}\right\} \tag{2}$$

where $BO_{ij}$ is the bond order between atoms $i$ and $j$, $r_{ij}$ is the interatomic distance, $r_o$ is the equilibrium bond length, and $P_{bo}$ is the empirical parameter. In Equation (2), the first, second, and third exponential terms determine the bond order contributions of single bond (σ-bond), double bond (π-bond), and triple bond (double π-bond). Each bonding term, $p$, and each bonding equilibrium distance, $r_o$, are parameterized to yield bond strengths and distances consistent with quantum mechanical predictions for species separated by distance $r_{ij}$ [51,53,65–67], as shown in Figure 2, where two carbon atoms are considered. The equilibrium distance, $r_o$, is about 0.154 nm. This results in a maximum bond order level of 3 for C–C. For C–H and H–H bonds, only the σ-bond contribution is considered, resulting in a maximum bond order level of 1. All bond-order functions fall smoothly to zero.

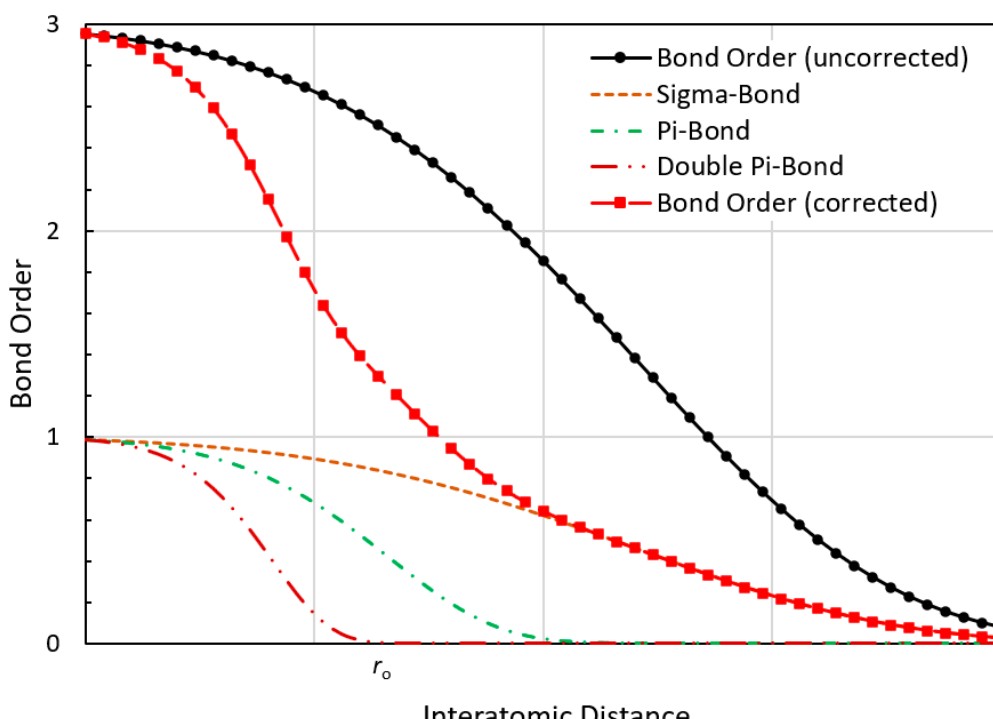

**Figure 2.** Schematic representations of uncorrected and corrected bond order in terms of interatomic distance vs. carbon–carbon bond order to be considered.

The classical MD simulations use force field potentials to describe the time evolution of any particle (atoms, molecules, or radicals) for a set of initial conditions based on Newton's second law (Newton's equations of motion). For atoms and/or molecular systems, the force field potentials (as uncorrected bond order in Figure 2) are calculated for each pair of atoms in the simulation as functions of the distance between atoms only. Solving Newton's equations of motion for a system constitutes the central task of classical MD, just as solving the Schrödinger equation is the central task of QM. However, in classical MD, subatomic electronic structures and dynamics are not calculated, thus excluding intrinsic QM events such as chemical reactions (i.e., chemical bond breaking and formation) and charge transfer. Whereas reactive MD seeks to balance power (chemical reactivity), accuracy, and computational cost. The reactive force field, particularly the ReaxFF method, expresses the energy terms (Equation (1)) as functions of bond order (or bond distance), such as σ-bond, π-bond, double π-bond, and the summation of the three bond orders—corrected bond order, as shown in Figure 2). The energy functions create a continuous surface connecting the reactants and produce and are generated through all intermediate states, which makes ReaxFF useful for simulating chemical reactions [50]. Based on the comparison of the schematic representations of the bond orders by the classical MD (uncorrected in Figure 2) and the ReaxFF MD (corrected in Figure 2), the difference, especially around the equilibrium bond distance, has been clearly shown.

### 2.3. Atomic Over-Coordination

According to the bonding valence theory, the total bond order of C should not exceed 4, and that of H should not exceed 1, except in the case of hyper valence. However, even after correcting the original bond orders $BO_{ij}$, some degree of over-coordination may still remain in the molecule. To fix this, an over-coordination penalty term has been added to the force field.

For over-coordinated atoms (the degree to which the sum of the uncorrected bond orders around the atomic center deviates from its valence $Val_i$, $\Delta_i > 0$), Equation (3) imposes an energy penalty on the system. The form of Equation (3) ensures that $E_{over}$ quickly

vanishes to zero for under-coordinated systems ($\Delta_i < 0$). $\Delta_i$ is calculated using Equation (4), using the corrected bond orders $BO_{ij}$ instead of the uncorrected bond orders.

$$E_{over} = p_{over} \cdot \Delta_i \cdot \left( \frac{1}{1 + exp(\lambda_1 \cdot \Delta_i)} \right) \tag{3}$$

$$\Delta_i = \sum_{j=1}^{nbond} BO_{ij} - Val_i \tag{4}$$

### 2.4. Valence Angle and Torsional Angle Interaction Terms

One of the disadvantages of the non-reactive force fields is their strict description of the angular and torsional interactions between atoms within the simulation. These types of interactions are often described as simple harmonic relationships, and the same harmonic potentials apply no matter how strong or weak the bond is. However, in ReaxFF, these angular and torsional interactions are also bond order dependent. This means that when an atom breaks a bond and leaves the molecule, the forces exerted on it due to the angle and torsion relative to the rest of the molecule diminish smoothly with the bond order. This is the expression in Equation (5):

$$E_{angle} = \left[ 1 - exp\left( \lambda \cdot BO_1^3 \right) \right] \left[ 1 - exp\left( \lambda \cdot BO_2^3 \right) \right] \cdot \left\{ k_a - k_b \cdot exp\left[ -k_b \cdot (\phi - \phi_o)^2 \right] \right\} \tag{5}$$

where $BO_1$ and $BO_2$ are the bond orders of each of the two bonds connecting the three atoms within the angle, $\lambda$ is an angle parameter set to obtain agreement with quantum values, $k_a$ and $k_b$ are the harmonic force constants determining the depth and width of the angular potential, respectively, $\phi$ is the angle, and $\phi_o$ is the equilibrium angle.

### 2.5. Coulombic and van der Waals Interaction Terms

The ReaxFF is also able to calculate the polarization of intramolecular charges. This is achieved by using the electronegativity and hardness parameters of each element in the system. These values have also been optimized using QM data. Equation (6) shows how to calculate the polarization:

$$\frac{\partial E}{\partial q_n} = \chi_n + 2 \cdot q_n \cdot \eta_n + C \cdot \sum_{j=1}^{n} \frac{q_j}{\left[ r_{n,j}^3 + (1/\gamma_{n,j})^3 \right]^{1/3}}, \quad \sum_{i=1}^{n} q_i = 0 \tag{6}$$

where $\chi_n$ is the electronegativity of element $n$, $\eta_n$ is the hardness of element $n$, and $\gamma_{n,j}$ is the shielding parameter between atoms $n$ and $j$ [51,53,65–67]. This method is based on the electronegativity equalization method (EEM) [53,68–70]. These charge values are determined for each time step of the simulation and depend on the geometry of the system.

Due to the rigid connectivity associated with non-reactive force fields, Coulomb and van der Waals forces are usually only calculated between pairs of atoms that do not share bonds or valence angles with each other. However, in the ReaxFF, the Coulomb and van der Waals forces are calculated between all pairs of atoms regardless of their connectivity [51,53,65–67]. To avoid excessively repulsive or attractive non-bonded interactions at short distances, Coulomb and van der Waals interactions are shielded in ReaxFF. This is achieved by using a shielding term, $\gamma$, as shown in the Coulombic calculation of Equation (7):

$$E_{Coulomb} = C \cdot \left\{ \frac{q_i \cdot q_j}{\left[ r_{ij}^3 + (1/\gamma_{ij})^3 \right]^{1/3}} \right\} \tag{7}$$

where $q_i$ and $q_j$ are the charges of the two atoms, $\gamma_{ij}$ is the shielding parameter, and $C$ is the coefficient. Atomic charges are calculated using the EEM approach [53,68–70]. The EEM charge derivation method is similar to the charge-equilibrium (QEq) scheme [70],

with the only difference, aside from the parameter definition, that EEM does not use an iterative scheme for hydrogen charges (such as QEq), while QEq uses a stricter Slater orbital approach to account for charge overlap. However, the $\gamma_{ij}$ in Equation (7) can be optimized to reproduce the QEq orbital overlap correction [51].

In addition to overlapping-dependent valence interactions, there are also repulsive interactions at short interatomic distances due to orthogonalization of the Pauli principle and attractive energies at long distances due to dispersion. These interactions, consisting of van der Waals and Coulomb forces, are included in all atomic pairs, thus avoiding awkward changes in the energy description during bond dissociation. In this respect, ReaxFF is similar in nature to the central valence force fields used earlier in vibrational spectroscopy [37]. To account for the van der Waals interactions, a distance-corrected Morse potential is used (Equations (8) and (9)). Excessively high repulsive forces between bonded atoms (1–2 interactions) and valence-sharing atoms (1–3 interactions) can be avoided by including shielding interactions (Equation (9)) [51,67].

$$E_{vdWaals} = T_{ap} \cdot D_{ij} \cdot \left\{ \exp\left[\alpha_{ij}\cdot\left(1 - \frac{f_{13}(r_{ij})}{r_{vdW}}\right)\right] - 2\cdot\exp\left[\frac{1}{2}\cdot\alpha_{ij}\cdot\left(1 - \frac{f_{13}(r_{ij})}{r_{vdW}}\right)\right] \right\} \quad (8)$$

$$f_{13}(r_{ij}) = \left[r_{ij}^{p_{vdW1}} + \left(\frac{1}{\gamma_{vdw}}\right)^{p_{vdW1}}\right]^{1/p_{vdW1}} \quad (9)$$

where

$$T_{ap} = \frac{20}{R_{cut}^7}\cdot r_{ij}^7 - \frac{70}{R_{cut}^6}\cdot r_{ij}^6 + \frac{84}{R_{cut}^5}\cdot r_{ij}^5 - \frac{35}{R_{cut}^4}\cdot r_{ij}^4 + 1 \quad (10)$$

where the terms in Equation (10) are used to avoid energy discontinuities outside the non-bonded cut-off radius ($R_{cut} = 1.0$ nm) of the ReaxFF. The terms in this polynomial are chosen to ensure that the first, second, and third derivatives of the non-bonded interactions with respect to the distance are all continuous and zero at the cut-off boundary. Therefore, this force field can adequately describe the weak van der Waals interactions even in the long range up to 1.0 nm.

*2.6. Conjugated System Term*

In theoretical chemistry, a conjugated system is a system in which p-orbitals of delocalized electrons are linked in a molecule, usually reducing the total energy of the molecule and increasing its stability. It is usually represented as having alternated single and multiple bonds. Lone pairs, radicals, or carbonium ions may be part of the system, which may be cyclic, linear, or mixed [71]. A conjugated system term is the overlap of one p-orbital with another p-orbital at an adjacent σ bond [72]. Molecules that contain conjugated systems of orbitals and electrons are called conjugated molecules, which have overlapping p-orbitals on three or more atoms. Some simple organic conjugated molecules are 1,3, butadiene, benzene, and allylic carbocations [73,74]. The largest conjugated systems are found in graphene, graphite, conductive polymers, and carbon nanotubes.

Equations (11) and (12) describe the contribution of conjugation effects to molecular energy. The conjugation energy contribution is greatest when the bond-order value of consecutive bonds is 1.5 (as in benzene and other aromatic compounds) [51].

$$E_{conj} = f_{12}\left(BO_{ij}, BO_{jk}, BO_{kl}\right)\cdot\lambda_{26}\cdot\left[1 + \left(\cos^2\omega_{ijkl} - 1\right)\cdot\sin\Theta_{ijk}\cdot\sin\Theta_{jkl}\right] \quad (11)$$

$$\begin{aligned} f_{12}&\left(BO_{ij}, BO_{jk}, BO_{kl}\right) \\ &= \exp\left[-\lambda_{27}\cdot\left(BO_{ij} - 1\tfrac{1}{2}\right)^2\right] \\ &\cdot\exp\left[-\lambda_{27}\cdot\left(BO_{jk} - 1\tfrac{1}{2}\right)^2\right]\cdot\exp\left[-\lambda_{27}\cdot\left(BO_{kl} - 1\tfrac{1}{2}\right)^2\right] \end{aligned} \quad (12)$$

### 2.7. Force Field Optimization Procedure

The force fields of reactive MD are optimized using a sequential single-parameter search technique as described by van Duin et al. [75]. In general, the search technique aims to reproduce the heat of formation to within 4.0 kcal/mol. Bond lengths are within 0.001 nm, and bond angles are within 2° of their literature values. In order to use the QC data during the force field optimization, structures related to these data are added to the force field training set. During the force field optimization, the energies of all molecules used to form the heat and geometric data comparisons are continuously minimized, while the structures related to the QC data are kept fixed or optimized with appropriate bond length or torsional angle constraints.

## 3. Results and Discussion

For all the above reasons, computational MD methods for biochar offer an alternative approach to understanding the mechanisms of biomass pyrolysis processes and carbonization reactions, predicting the outcomes and compositions of biochar after the processes, and characterizing the performance and functionalities of the biochar without costing a lot and taking a lot of time to produce, characterize, and experimentally study the insight into the final biochar and carbon materials. Furthermore, computational methods can be successfully employed to screen better biomass feedstocks or materials with suitable properties early in the development of various processes, which can provide opportunities for further research and evaluation and provide complementary information to experimentally collected data.

### 3.1. The Compositions and Physicochemical Properties of Biomass Feedstocks and the Produced Biochar

As mentioned above, biochar can be produced through thermochemical processes (e.g., torrefaction, pyrolysis, gasification, combustion, etc.). Biomass pyrolysis is generally defined as the thermal decomposition of biomass organic matrix in a non-oxidizing atmosphere to produce liquid bio-oil, solid biochar, and non-condensable gas products (also named syngas) [76]. Therefore, pyrolysis is demonstrated as one of the most promising technologies for converting biomass into biochar [77]. Biomass, like forest wastes and agricultural residues, is the most common feedstock for biochar production, which mainly consists of cellulose, hemicellulose, and lignin. The contents of cellulose, hemicellulose, and lignin vary significantly depending on the feedstock species, parts (leaf, root, stem, etc.) of plants, and growth conditions. The cellulose content may range from 40–60%, the hemicellulose content is between 15 and 30%, and the lignin content may vary from 10–25%. In addition to these three major components, a small fraction of extracts and inorganic ash are also present in biomass feedstocks as non-structural components that do not constitute the cell walls or cell layers [78]. Agricultural residues and forest wastes usually contain carbon (C) (40–55%), oxygen (O) (30–45%), hydrogen (H) (5–10%), nitrogen (N) (0.5–3%), sulfur (S) (<1%), and other trace elements [7–9], which may result in significant variations in the properties and compositions of produced biochar. For example, biochar is produced from cassava rhizomes, cassava stalks, and corncobs using a patented kiln by farmers. It was found that the biochar derived from cassava stems yielded the highest BET surface area among the biochar products, while the biochar produced from corncobs yielded the highest C (81.35%), and highest H content (2.42%). This was because the corncobs contain the highest C content (41.66%), the highest H content (6.84%), the lowest N content (0.74%), and the lowest O content (50.76%) [79]. In contrast, the cassava rhizomes contained the lowest C content (37.60%), the lowest H content (6.04%), the highest N content (1.27%), and the highest O content (55.3%). The study also showed that the biochar produced by slow pyrolysis with a longer reaction time at a lower heating rate and temperature was of high quality, stable C, and had a low H/C ratio. The high BET surface area and total pore volume of this biochar make it suitable for soil amendment, helping to reduce soil density, improve soil moisture and aeration, and reduce the leaching of plant nutrients from the rhizosphere.

Pyrolysis models usually pre-consider the reactions of each individual component of biomass without considering the potential interactions between the components. The behavior of biomass pyrolysis can be considered a comprehensive performance of the thermal cracking behavior of the three main components (cellulose, hemicellulose, and lignin). Table 1 lists the physicochemical properties, and Table 2 shows the major elemental compositions of different biochar samples from the example above [79–84]. All these data were used for the MD simulation model generation.

**Table 1.** Major physicochemical properties of biochar [80].

| Pyrolysis Type | SSA ($m^2/g$) | CEC (cmol/kg) | AEC (cmol/kg) | CCE (%) | PV ($m^3/t$) | APS (nm) | Ash (%) | pH | EC (dS/m) |
|---|---|---|---|---|---|---|---|---|---|
| Fast | 183 ± 17.3 | 44.9 ± 3.62 | 4.90 ± 3.45 | 6.10 ± 1.12 | 2.04 ± 0.81 | 52.3 ± 40.2 | 19.2 ± 19.2 | 8.7 ± 0.1 | 4.43 ± 0.50 |
| Slow | 98.6 ± 3.53 | 48.1 ± 3.12 | 5.33 ± 1.51 | 11.2 ± 0.98 | 3.66 ± 1.27 | 1190 ± 565 | 22.0 ± 0.51 | 8.7 ± 0.0 | 5.85 ± 1.58 |
| Feedstock source | SSA ($m^2/g$) | CEC (cmol/kg) | AEC (cmol/kg) | CCE (%) | PV ($m^3/t$) | APS (nm) | Ash (%) | pH | EC (dS/m) |
| Wood based | 184 ± 11.4 | 23.9 ± 1.87 | 5.65 ± 1.80 | 9.04 ± 1.17 | 7.01 ± 3.07 | 74.6 ± 44.4 | 10.2 ± 0.43 | 8.3 ± 0.1 | 6.20 ± 2.85 |
| Crop wastes | 98.2 ± 5.45 | 56.3 ± 3.92 | 4.51 ± 1.96 | 6.12 ± 0.97 | 2.05 ± 0.91 | 2320 ± 1150 | 21.1 ± 0.54 | 8.9 ± 0.1 | 5.72 ± 0.67 |
| Other grasses | 63.4 ± 8.84 | 63.316.4 | 2.05 ± 1.05 | — | 3.36 ± 3.30 | 268 ± 125 | 18.0 ± 1.01 | 8.9 ± 0.1 | 5.20 ± 0.93 |
| Manures/biosolids | 52.2 ± 4.23 | 66.1 ± 8.00 | 7.77 ± 7.52 | 14.2 ± 1.56 | 0.82 ± 0.30 | 27.3 ± 12.5 | 44.6 ± 0.97 | 8.9 ± 0.1 | 3.98 ± 0.41 |

SSA—specific surface area; CEC—cation exchange capacity; AEC—anion exchange capacity; CCE—calcium carbonate equivalent; PV—total pore volume; APS—average particle size; EC—electrical conductivity.

**Table 2.** The major elemental compositions of biochar [79,80].

| Pyrolysis Type | C (wt.%) | H (wt.%) | O (wt.%) | N (wt.%) | S (wt.%) |
|---|---|---|---|---|---|
| Fast | 60.6 ± 0.47 | 3.37 ± 0.08 | 19.1 ± 0.38 | 1.63 ± 0.06 | 0.085 ± 0.009 |
| Slow | 60.8 ± 0.34 | 3.36 ± 0.09 | 18.4 ± 0.29 | 1.63 ± 0.04 | 0.055 ± 0.004 |
| Feedstock source | C (wt.%) | H (wt.%) | O (wt.%) | N (wt.%) | S (wt.%) |
| Wood based | 70.5 ± 0.39 | 3.38 ± 0.08 | 17.7 ± 0.35 | 0.95 ± 0.03 | 0.044 ± 0.007 |
| Crop wastes | 61.4 ± 0.41 | 3.28 ± 0.10 | 18.1 ± 0.38 | 1.54 ± 0.06 | 0.039 ± 0.006 |
| Other grasses | 63.6 ± 0.72 | 5.11 ± 0.50 | 20.9 ± 0.74 | 1.80 ± 0.14 | 0.051 ± 0.021 |
| Manures/biosolids | 41.6 ± 0.68 | 2.73 ± 0.10 | 16.5 ± 0.70 | 2.42 ± 0.06 | 0.089 ± 0.006 |
| Corncobs/cassava rhizomes/cassava stems | 62.95–81.35 | 2.24–2.73 | 15.23–33.44 | 1.22–1.65 | — |

### 3.2. Carbonization Reactions in Biomass Pyrolysis Processes

MD simulations based on ReaxFF can be a powerful tool to analyze the biomass pyrolysis process. By adopting the bond-order formalism in a classical approach, ReaxFF implicitly describes the chemical bonds without costly QM calculations, giving insight into the biomass pyrolysis process and its carbonization. It can describe the bond breaking and formation during the chemical reactions, thereby exploring the complex carbonization reaction mechanisms at a nano/microscale, including different pyrolysis processes of various feedstocks as well as other chemical process mechanisms for producing biochar or other carbonaceous materials [29,30,52,55,59,64,76–78,85–100], such as thermochemical reactions in combustion and energy systems [50,53,57,101], energetic and dissociative water properties under various conditions [56], properties of carbon nano-rings, carbon nanotube bundles, and crosslinked epoxy resins [66,67,102], inclusion of geometry-dependent charge calculations [75], to name a few. Biomass can produce biochar through thermochemical processes such as pyrolysis, gasification, and combustion [77]. Chen et al. [57] constructed a simplified biomass model containing all three main components of cellulose, lignin, and hemicellulose and applied it to the simulations of pyrolysis and combustion processes under various oxidative and humid environments. The pyrolysis models usually presuppose

the reactions of each individual component without considering the potential interactions between the major components.

### 3.2.1. Reactivity of Cellulose in Pyrolysis

Cellulose is the most abundant organic compound on earth. Cellulose, with the chemical formula $(C_6H_{10}O_5)_n$, is a polysaccharide composed of linear chains of hundreds to tens of thousands of 1,4-β-D-glucopyranose units. The schematic diagram of the long linear chain molecule structure of cellulose is shown in Figure 3.

**Figure 3.** Schematic structure of cellulose $(C_6H_{10}O_5)_n$.

By weight, cellulose accounts for 40–50% of biomass. It is difficult to understand the complex reaction process and detailed reaction mechanism of biomass pyrolysis only through experimental methods. Studying pyrolysis mechanisms and the product pathway of cellulose is of great significance for producing new carbon materials and exploring innovative applications. When using pyrolysis–gas chromatography–mass spectrometry (Py-GC/MS) to study the pyrolysis mechanism of cellulose, since Py-GC/MS cannot capture the reactions related to free radicals at the molecular level in a short time, it cannot provide detailed information on the pyrolysis mechanism [78,103]. ReaxFF MD simulations are effective methods to reveal the internal reaction mechanisms from a microscopic perspective. The decomposition of cellulose was classified into three categories using ReaxFF MD [104]: (1) Depolymerization reactions; (2) other chain scission reactions; and (3) release of low molecular weight products such as glycolaldehyde, water, formaldehyde, and formic acid. Since the simulated temperature is higher than the experimental temperature and the chemical and electrostatic ambient temperature of crystalline (or amorphous) cellulose does not exist, levoglucosan (LGA) is not observed in the simulation, and the kinetic parameters do not depend on molecular weight or initial conformation. Then, post-processing tools were used to parse the bond information. Specific algorithms were developed to search for LGA among decomposition products, taking into account both individual molecules and LGA end groups. Similar studies have been performed on glycolaldehyde, formaldehyde, formic acid, and hydroxymethyl radical. Some smaller products, such as water, carbon monoxide, carbon dioxide, and hydroxyl radicals, can be identified by their chemical compositions.

It should be pointed out that the experimental techniques, including product-specific Py-GC/MS experiments, cannot directly monitor the temporal evolution trend of specific products during cellulose pyrolysis because the radial reaction time is many orders of magnitude shorter than the experimental techniques allow, resulting in a lower concentration of pyrolyzed products. However, the simplicity of observing the evolution of different products as a function of time and temperature through ReaxFF MD simulations provides a feasible method for computationally probing the evolution of the pyrolysis products for experimental or industrial applications [97].

Zhang, et al. [97] combined large-scale models with GPU-based ReaxFF MD simulations using a canonical ensemble (conservation of substance quantity (N), volume (V), and temperature (T) of species, also known as the NVT ensemble) in a periodic cubic box and a unique cheminformatics-based reaction analysis tool (VARxMD) and studied the cellulose pyrolysis process and revealed the evolution and reaction mechanism of cellulose

pyrolysis products. The three major products are hydroxyl-acetone, propyl aldehyde, and glycolaldehyde, in addition to levoglucosan, $CO_2$, CO, $H_2O$, etc., which are closely related to the experimental literature. Both the overall spectral product evolution and the underlying detailed chemical reactions of cellulose pyrolysis have been revealed. For example, the reaction pathway of hydroxyl-acetone is to undergo a series of depolymerization reactions such as homolysis, elimination, and ring opening at 800K to form the unstable compound $C_6H_{10}O_6$. The unstable $C_6H_{10}O_6$ then undergoes ring formation, rearrangement of unsaturated bonds, and hydroxyl radicals falling off to generate fragment $C_6H_9O_5$. $C_6H_9O_5$ decomposes into two species by bond dissociation, namely 2-hydroxyl-malonaldehyde ($C_3H_4O_3$) and 2-hydroxyl-propyl-aldehyde radical ($C_3H_5O_2$). 2-hydroxyl-propyl-aldehyde radial rearranges and reacts with the hydrogen radical to generate hydroxyl-aceton $C_3H_6O_2$. In the reaction pathway of glycolaldehyde, $C_2H_4O_3$, at 800 K, the generation of $C_2H_4O_3$ comes from the bond dissociation of different compounds, such as $C_6H_{10}O_6$ and $C_{11}H_{19}O_9$ released by depolymerization of cellulose [97]. Since $C_6H_{10}O_6$ is the initial reactant for the formation of hydroxyl-acetone and 2-hydroxy-propionaldehyde, the same initial fragment may undergo different reactions in the cellulose pyrolysis system to produce different compounds, and the three main products compete with each other for formation at high temperatures. The weight percentage of the pyrolysis products was found to be a function of temperature. When the temperature is low, the decomposition process predominates. As the temperature increases, the rate of thermal degradation of cellulose is accelerated, accompanied by the appearance of fragments and inorganic gas molecules. The inorganic gas production also increases with temperature between 500 and 1400 K, which is consistent with Py-GC/MS experiments. The pyrolysis products can be grouped roughly into syngas, tar (or bio-oil), and biochar [104]. Syngas and its gas compounds have carbon atoms equal to or less than 4, and bio-oil is a collection of compounds of $C_5$–$C_9$, $C_{10}$–$C_{19}$, $C_{20}$–$C_{29}$, and $C_{30}$–$C_{39}$. The rest ($C_{40+}$) can be treated as biochar. The gases in syngas are released rapidly and increase monotonically with increasing temperature. When the temperature is increased, biochar is rapidly reduced, so a lower temperature favors a higher yield of biochar. The above results are in good agreement with the experimental data [105]. Analysis of the molecular products formed by cellulose pyrolysis in ReaxFF MD simulations at different time steps and temperatures ranging from 800K to 1400 K revealed that two compounds, namely $C_2H_4O_2$ and $C_6H_{10}O_5$, dominated the system. With the help of VARxMD, it was found that most of $C_2H_4O_2$ is glycolaldehyde and $C_6H_{10}O_5$ is levoglucosan or the precursor of levoglucosan, which is consistent with the experimental data of Py-GC/MS. At 700 K, the precursor of levoglucosan constitutes about 80% of the $C_6H_{10}O_5$ compounds, which drops to only 15% at 1400 K. Since levoglucosan is a large molecule for the cellulose model, both levoglucosan and its precursors are considered levoglucosan products. The evolution trend of the large model simulation is closer to that of the Py-GC/MS experiments than the small model [97].

### 3.2.2. Reactivity of Lignin in Pyrolysis

Lignin is a polyphenolic polymer. Three types of phenylpropanoid units are generally considered the main precursors for lignin biosynthesis: coniferyl, sinapyl, and p-coumaryl alcohol (see Figure 4), which structurally give rise to guaiacyl (G), syringyl (S), and p-hydroxyphenyl (H) units, respectively, and are linked by different C–C bonds and ether bonds, such as α-O-4 bonds and β-O-4 bonds [106]. Lignin is the most recalcitrant of the three components of lignocellulosic biomass. Lignin, the second most abundant natural polymer after cellulose, is pyrolyzed differently from cellulose and hemicellulose due to the differences in chemical structure and characteristics [76]. Generally speaking, the lignin in softwoods is mainly composed of guaiacyl units and contains a small number of p-hydroxyphenyl units; in contrast, the lignin in hardwoods is mainly composed of guaiacyl units and syringyl units and contains a small number of p-hyfroxyphenyl units. The lignin in grasses typically contains all three types of monlignol units, with peripheral groups (i.e., hydroxycinnamic acids) incorporated into its core structure. The lignin macromolecules

are mainly linked by C–C and C–O bonds between their phenylpropanoid structural units, among which aryl ether bonds (β-O-4) are the most common and important interunit linkages (see Figure 5).

**Figure 4.** Schematic basic units of lignin polymer: (**a**) coniferyl, (**b**) sinapyl, and (**c**) p-coumaryl alcohol structures.

**Figure 5.** Schematic structure of lignin, where the purple wavy lines represent the continuing of the molecule.

It can be seen from Figure 5 that lignin contains a variety of oxygen-containing functional groups, including methoxyl, hydroxyl, carboxyl, and carbonyl groups, etc., which significantly affect the reactivity of lignin. The content of methoxyl groups in lignin is related to the formation of lignin pyrolysis char, i.e., lignin with high methoxyl group contents produces less char during the pyrolysis process.

Many lignin models have been proposed for ReaxFF MD simulations in the last century, including the softwood lignin models by Alder, Freudenberg, Brukraft, Forss, and Glasser; the hardwood lignin model by Nimz; and the pine kraft lignin model by Marton [76,107]. The corresponding details of the four lignin models are listed in Table 3 [76,107].

**Table 3.** Constituent details of four lignin models [76,107].

| Lignin Model | Species | Constituents | Number of $C_6H_3$ Units | | |
| --- | --- | --- | --- | --- | --- |
| | | | H | G | S |
| Alder | Softwood | $C_{6400}H_{7200}O_{2320}$ | 40 | 560 | 40 |
| Freudenberg | Softwood | $C_{6980}H_{7640}O_{2280}$ | 200 | 480 | 40 |
| Nimz | Hardwood | $C_{10854}H_{11940}O_{4062}$ | 42 | 612 | 396 |
| Marton | Kraft lignin | $C_{5080}H_{5040}O_{1640}S_{40}$ | 120 | 440 | 0 |

The simulation results show that the Alder, Frendenberg, and Nimz lignin models have three pyrolysis stages. The first stage refers to the main process, starting with the formation of initial pyrolysis products and ending with the complete consumption of source lignin molecules. The second stage starts with the cracking of primary pyrolysis products into secondary pyrolysis products and ends with a maximum. The third stage begins with the reduction of secondary pyrolysis products. However, the Marton lignin model only has the first and second stages. Consumptions of all α-O-4 linkages were similar in the softwood and hardwood lignin models. The differences in pyrolysis product evolution and linkage behavior between hardwood, softwood, and kraft lignin can be attributed to the different reactions of linkages and their linked monomers induced by different oxygen-containing substituents [76]. The ring structure evolutions of the four lignin models are almost identical. The similarities and differences in the pyrolysis of the different lignin models suggested that the simulation work can provide new insights into the high-value utilization of lignin.

The study of the ReaxFF MD simulated pyrolysis of lignin model will be helpful for better understanding the reaction behavior and pyrolysis characteristics in the pyrolysis of lignin, so as to optimize the pyrolysis process of lignin, and even the whole biomass feedstock. Furthermore, lignin is heterogeneous and does not have a well-defined primary structure. The phenylpropane units are organized into a 3D amorphous polymeric network with varying degrees of aggregation. Abundant forms of biomass are a promising alternative to fossil fuels [108]. Lignin depolymerization is a difficult process that requires specific reaction conditions [109,110] and selected catalysts [108] to increase the yield of high aromatic monomers and produce value-added fuels and chemicals.

3.2.3. Reactivity of Hemicellulose in Pyrolysis

Hemicellulose has a heteropolymeric structure (lower molecular weight than cellulose) that is composed of a variety of sugar monomers, including glucose, galactose, mannose, xylose, arabinose, 4-O-methyl glucuronic acid, and galacturounic acid residues. The composition of hemicellulose from different biomass materials varies, and its structure is more complex than that of cellulose. Xylan is often used as a typical hemicellulose model [111,112], as shown in Figure 6. Previous work has provided important insights into the relationship between the distribution of pyrolysis products and the structural features of xylosyl hemicelluloses [113–116]. The main differences between xylan and cellulose pyrolysis are as follows: (1) xylan melts and generates bubbles during pyrolysis, producing

many unidentified didehydrated pentose, while levoglucosan is the main compound in the dellulose bio-oil; (2) xylan and pyranose molecules formed during the pyrolysis of xylan tend to form char through multi-step dehydration reactions [113–120].

**Figure 6.** Schematic structure of xylan, where the purple wavy lines represent the continuing of the molecule.

Furthermore, ReaxFF MD simulations showed that cellulose and hemicellulose were the main sources of CO and $CO_2$ production, although there were slightly more $CO_2$ molecules in hemicellulose due to the potential presence of carboxyl and carbonyl groups. Due to the high O content, cellulose and hemicellulose are the main sources of $C_2H_2O_2$ and $CH_3CHO$. Most of the $C_2H_4O_2$ molecules come from the degeneration of the cellulose. In addition, $CH_3OH$ is mainly derived from cellulose due to the cleavage of the hydroxyl-rich pyran rings. The dissociation of $-OCH_3$ radicals in the lignin also provides the precursors for the formation of $CH_3OH$ [59]. Table 4 lists the percentage of each gas produced during the pyrolysis process of the three main components of the biomass (wheat straw) at a temperature of 2000 K [59].

**Table 4.** Percentage of each gas produced resulting from the three main components of the biomass (wheat straw) during the pyrolysis process at 2000 K.

|  | CO (%) | $CO_2$ (%) | $CH_4$ (%) | $H_2O$ (%) |
|---|---|---|---|---|
| Cellulose | 48.29 | 47.04 | 53.33 | 83.96 |
| Lignin | 7.32 | 4.68 | 40.00 | 5.30 |
| Hemicellulose | 44.39 | 48.28 | 6.67 | 10.74 |

Functional groups play an important role in the pyrolysis process and determine the chemical properties of organic compounds. Cellulose, hemicellulose, and lignin in biomass have different kinds and numbers of functional groups, which lead to their different properties. In addition, phenyl rings in lignin and pyran rings in cellulose and hemicellulose also have potential effects on the thermal stability of organics. Therefore, the study of its evolution and behavior is beneficial to understanding the mechanism of biomass pyrolysis [59]. The functional groups are generally divided into six categories: ether groups (R–O–R), hydroxyl groups (R–OH), aldehyde groups (R–CHO), ester groups (R–COO–R), carboxyl groups (R–COOH), and carbonyl groups (R–CO–R). The ether and hydroxyl groups are the most abundant, while the other four functional groups are much less abundant. The hydroxyl groups are concentrated in the cellulose and hemicellulose in the form of alcoholic hydroxyl groups [121]. The ether groups are mainly derived from the pyran rings and glycosidic bonds in the cellulose and hemicellulose, as well as the methoxy or other carbon structures connected to the phenyl rings in the lignin [122]. The aldehyde

and carbonyl groups are concentrated in lignin, while hemicellulose provides carboxyl groups. Furthermore, the ester groups are distributed in hemicellulose and lignin.

### 3.3. Additional Notes

While computational methods are inexpensive, they can sometimes provide insight information where experimental methods are impossible. However, computational methods have their own limitations. In general, any model is limited by the size of the structure being modeled, the period of reaction time it can simulate, and ultimately the processing power available. QM-based ab initio and DFT can provide more accurate representations of materials by calculating the potentials acting on all atoms from first principles, but they are computationally resource intensive, limiting their usefulness for predicting the behavior of bulk materials.

Classical MD approximates the interatomic potentials using force fields that have been parameterized using ab initio calculations and empirical data to approximate the behavior of the system as accurately as possible, with well-defined bonds, angles, dihedrals, and impropers explicitly.

ReaxFF is a reactive implementation of MD where bonding is implicit and the bond orders between atomic pairs are functions of interatomic distances. By setting the parameters of hydrocarbon reactions or carbonization processes, reactive MD has been applied to the combustion, oxidation, and pyrolysis of coal [57,90–93], the pyrolysis reaction processes and mechanisms of polyethylene [94], and polycarbonate [95,96], to name a few. Among other things, modeling the thermal decomposition of cured epoxy resins and their mechanical responses elucidated unique insights into decomposition steps and failure modes, which were difficult to determine experimentally [98]. Furthermore, studies of the pyrolysis of polymeric ablative materials for thermal protection systems, a popular choice of ablator using ReaxFF, are another application area [102]. The potential applications of reactive MD simulations can be extended to processes involving chemical reactions, such as the design and evaluation of ablative materials, the design and development of energetic materials, the design of hydrogen storage materials, the study of petrochemical manufacturing methods, etc.

Many key factors, such as the biomass feedstock species, reactor types and operating conditions, post-treatment of biochar activation, physiochemistry, morphology, performance, and functionalities of the biomass, synergistically affect the pyrolysis and carbonization processes and the production of inorganic gas, organic gas, biochar, bio-oil, etc. [29,30,44,57,111,113]. The reactive processes and mechanisms that can be simulated and analyzed quantitatively and accurately remain challenging.

## 4. Conclusions

Each biochar produced is unique due to the biomass feedstock species, the reactor types and operating conditions, and the synergistic effects of different factors. The biochar production processes are complex and involve chemical reactions that determine the physicochemical properties of the final biochar produced. Classical MD simulations cannot effectively analyze the mechanisms of biomass pyrolysis processes and carbonization reactions This paper has reviewed the fundamentals of the ReaxFF method and the corresponding MD simulation findings based on ReaxFF to characterize the biomass pyrolysis processes and the physicochemical properties of the produced biochar, as well as the carbonization reactions in pyrolysis. The possible composition of the biomass and the physicochemical properties of the produced biochar have been summarized. The models and reactivities of the three major components of biomass: Cellulose, lignin, and hemicellulose in pyrolysis have been discussed. The potential applications of ReaxFF MD were briefly discussed. The MD simulations based on ReaxFF can be an effective method for understanding various processes involving chemical reactions, such as the design and evaluation of ablative materials, the design and development of energetic materials, the design of hydrogen storage materials, the investigation of petrochemical manufacturing methods, in addition to the carbonization mechanisms of the biomass pyrolysis process.

The information derived from the simulations will be helpful for further optimization of the processes.

**Author Contributions:** Conceptualization, Z.H. and L.W.; methodology, Z.H. and L.W.; formal analysis, Z.H.; investigation, Z.H.; resources, Z.H. and L.W.; data curation, Z.H.; writing—original draft preparation, Z.H.; writing—review and editing, Z.H. and L.W.; visualization, Z.H.; supervision, Z.H.; project administration, Z.H.; funding acquisition, Z.H. All authors have read and agreed to the published version of the manuscript.

**Funding:** This research received no external funding.

**Data Availability Statement:** Not applicable.

**Acknowledgments:** This work was supported by the Department of Mechanical Engineering in the J. J. Lohr College of Engineering and the Department of Agricultural & Biosystems Engineering in the College of Agriculture, Food & Environmental Sciences at South Dakota State University and are gratefully acknowledged.

**Conflicts of Interest:** The authors declare no conflict of interest. The funders had no role in the design of the study; in the collection, analyses, or interpretation of data; in the writing of the manuscript; or in the decision to publish the results.

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
