# Peer review of "Review on Characterization of Biochar Derived from Biomass Pyrolysis via Reactive Molecular Dynamics Simulations"

_jcs, doi:10.3390/jcs7090354_

Round 1

Reviewer 1 Report

The submitted paper (Manuscript ID: jcs-2561954, 'Review on Characterization of Biochar ...') is, primarily, a good review. It has a fine basic skeleton, but there is the following suggestion for more insights. It has, only, a limited consideration, and no deep learning context through clean-cut discussion and links (as illustrative representation or explanatory material pathways) among the basic theory approach (ReaxFF methods and/or simulations) with/and certain experimental implementations or other (simulation) performance cases. All the partial contributions to the total system energy in equation#1 (Line_223) should get more ('additional notes') discussion/notes including an integrated/characteristic link(s)/ref(s) with, at least one, of the presented (paradigmatic) archetypical figures (Figures 3,4,5, and 6).
Also, there are a few (additional) minor (11) comments/errors.

Minor (11) comments/errors.

1. Line_101; Add a term. Add the term 'effective', e.g. to be high effective surface, in: "porous carbons are responsible for the high surface area in these materials, it is crucial to".

2. Line_242; A (possible) typo error. Add the indeces 'ij' ('BO->BOij'), in: "where BO is the bond order between atoms i and j, rij is the interatomic distance, ro is the"

3. Line_358; A typo error. Substruct a letter 'f', ('offfer->offer'), in: "For all the above reasons, computational MD methods for biochar offfer an".

4. Line_370; A typo error. Correct the term 'thermalchemical', ('thermochemical' ?), in: "As mentioned above, biochar can be produced through thermalchemical processes".

5. Line_425; A typo error. Add the letter 's', ('variou ' -> 'various'), in: "processes of variou feedstocks, as well as other chemical process mechanisms for".

6. Line_464; A typo error. Correct the word 'induvidual' ( -> 'individual'), in: "taking into account both induvidual molecules and LGA-end groups. Similar studies have".

7. Line_478; A typo error. Correct the word 'temperatuer' ( -> 'temperature'), in: "(V), and temperatuer (T) of species, also known as NVT ensemble) in a periodic cubic box".

8. Line_482; Erratum. Homogenize/Correct the names 'hydroxyl-acetone', 'propyl aldehyde', in: "hydroxyl-acetone, 2-hydroxy propyl aldehyde and glyrolaldehyde, in addition to levoglucosan, CO2, Co, and H2O, etc.,".

9. Line_496; A typo error. Correct the word 'reactons' ( -> 'reactions'), in: "the same initial fragment may undergo different reactons in the cellulose pyrolysis system".

10. Line_576; Typo errors. Correct the words 'heterogeneroud', and 'definted', in: "feedstock. Furthermore, lignin is heterogeneroud and does not have a well-definted".

11. Line_628; A typo error. Correct the word 'concentratd' ( -> 'concentrated'), in: "and carbonyl groups are concentratd in lignin, while the hemicellulose provides carboxyl".

Author Response

Greatly appreciated you taking time to review our manuscript and provide valuable comments and suggestions. We have made revisions and corrections accordingly, please refer to the revised manuscript and the cover letter with point-to-point responses. Thanks.

Reviewer 2 Report

The manuscript covers the fundamentals of the ReaxFF method, pyrolysis of biomass and its carbonization reactions, including the reactivity of major biomass components such as cellulose, lignin and hemicellulose. The work is interesting, but some modifications are requested before its publication in JCS.

1. The abstract is too long.

2.Other cases can be added in the introduction, such as the use of hydrothermal carbonization of orange peels 10.3390/pharmaceutics14102249; or https://doi.org/10.1016/j.biortech.2015.02.035 and so on.

3. Figure 3: Improves the figure. The bonds must all be the same length.

4. Figure 4: See previous comment; the difference between the size of the benzene and the bond with the methoxy group is too disproportionate. Consider this comment for Figure 6 as well. 

5. No pictures on the simulations?

6. Correct the numerous typos in the manuscript.

7. Conclusions are too general. What are the future prospects?

Moderate editing of English language required

Author Response

(The authors gave the same response as above.)

Round 2

Reviewer 2 Report

Suitable corrections were made. I accept this manunuscript in present form.